# Hepatocellular Carcinoma: Molecular Pathogenesis and Therapeutic Advances

**DOI:** 10.3390/cancers14030621

**Published:** 2022-01-26

**Authors:** Blanca Cucarull, Anna Tutusaus, Patricia Rider, Tania Hernáez-Alsina, Carlos Cuño, Pablo García de Frutos, Anna Colell, Montserrat Marí, Albert Morales

**Affiliations:** 1Department of Cell Death and Proliferation, IIBB-CSIC, IDIBAPS, 08036 Barcelona, Spain; blanca.cucarull@iibb.csic.es (B.C.); anna.tutusaus@iibb.csic.es (A.T.); rider@clinic.cat (P.R.); carlos.cuno@iibb.csic.es (C.C.); pablo.garcia@iibb.csic.es (P.G.d.F.); anna.colell@iibb.csic.es (A.C.); 2Digestive Unit, Hospital San Pedro, Rioja Salud, 26006 Logroño, Spain; thernaez@riojasalud.es; 3Unidad Asociada (IMIM), IIBB-CSIC, CIBERCV, IDIBAPS, 08036 Barcelona, Spain; 4Network Center for Biomedical Research in Neurodegenerative Diseases (CIBERNED), 08036 Barcelona, Spain; 5Barcelona Clinic Liver Cancer Group, Liver Unit, Hospital Clínic of Barcelona, University of Barcelona, CIBEREHD, IDIBAPS, 08036 Barcelona, Spain

**Keywords:** liver cancer, molecular therapies, immune checkpoint inhibitors, tyrosine kinase inhibitors, tumor microenvironment

## Abstract

**Simple Summary:**

Patients with unresectable hepatocellular carcinoma (HCC), the most common primary tumor of the liver, have poor prognosis and are increasing worldwide. The recent approval of several novel therapies for HCC was long expected, and it will make physician decision-making more challenging. The molecular mechanisms triggered during chronic liver diseases and the cellular cross-talk established with liver cells influence HCC growth and may reduce immune control, making this knowledge relevant to help with clinical decisions. This review analyzes these issues and points to relevant topics for future research.

**Abstract:**

Hepatocellular carcinoma (HCC), the most common form of liver cancer, continues to be a serious medical problem with poor prognosis, without major therapeutic improvement for years and increasing incidence. Fortunately, advances in systemic treatment options are finally arriving for HCC patients. After a decade of sorafenib as a standard therapy for advanced HCC, several tyrosine kinase inhibitors (TKIs), antiangiogenic antibodies, and immune checkpoint inhibitors have reached the clinic. Although infections by hepatitis B virus and hepatitis C virus remain principal factors for HCC development, the rise of non- alcoholic steatohepatitis from diabetes mellitus or metabolic syndrome is impeding HCC decline. Knowledge of specific molecular mechanisms, based on the etiology and the HCC microenvironment that influence tumor growth and immune control, will be crucial for physician decision-making among a variety of drugs to prescribe. In addition, markers of treatment efficacy are needed to speed the movement of patients towards other potentially effective treatments. Consequently, research to provide scientific data for the evidence-based management of liver cancer is guaranteed in the coming years and discussed here.

## 1. Hepatocellular Carcinoma

### 1.1. Epidemiology

Liver cancer is the sixth-most-frequent neoplasm and the third-most-frequent cause of cancer-related death, with approximately 900,000 new cases and 830,000 deaths in 2020 [1,2]. Among liver cancers, hepatocellular carcinoma (HCC) accounts for around 75% of primary liver tumors [3]. HCC develops in a context of chronic liver disease, and in most cases incidence rates of HCC among men are 2- to 4-fold higher than rates among women [4]. Common risk factors are chronic infection with hepatitis B virus (HBV) and aflatoxin B1 exposure in eastern Asia and sub-Saharan Africa [5], while, in Europe, Japan, and North America, the main risk factors are hepatitis C virus (HCV) and alcohol use [6]. Unfortunately, the prevalence of metabolic risk factors for HCC, including metabolic syndrome, obesity, type II diabetes, and nonalcoholic fatty liver disease (NAFLD), are emerging as HCC causes and may jointly become the leading cause of HCC worldwide in the near future, while incidence due to HBV or HCV will likely decline [4]. These causes, together with tobacco and some dietary factors, such as high iron intake, also increase the risk of developing HCC [4].

The expected decline in virus-related HCC incidence is due to HBV vaccination programs, a successful public health strategy, and HCV treatment (interferon and direct-acting antivirals) that might reduce the risk of HCC development, particularly in patients with sustained viral response. Coffee consumption and statins use have also been linked to a decrease in HCC incidence [7,8].

### 1.2. Molecular Pathogenesis

#### 1.2.1. Cellular Origin

The cell of origin of HCC remains elusive, probably due to the heterogeneity of liver cancer within the same tumor and between different tumors [9]. Previous studies have suggested, as in numerous cancer types, that liver stem cells may be responsible for initiating HCC, but a transit-amplifying population or mature hepatocytes could also be responsible. Nonetheless, in contrast to most organs, the liver lacks a defined stem cell population for organ maintenance. Some preclinical murine models of HCC support the likelihood of mature hepatocytes, and not of progenitor cells, as the cellular source of HCC [10]. This transformation of hepatocytes can take place via a sequence of genetic alterations or through dedifferentiation into hepatocyte precursor cells, which then become HCC cells that express progenitor cell markers, or by transdifferentiation into biliary-like cells that give rise to intrahepatic cholangiocarcinoma [9], therefore suggesting extraordinary hepatocyte cell plasticity [11].

#### 1.2.2. Molecular Drivers

About 70–80% of HCC develop in a context of cirrhosis that involves a complex multistep process [6]. In the cirrhotic liver, HCC starts with the presence of pre-cancerous cirrhotic nodules, called low-grade dysplastic nodules (LGDNs), that can transform into high-grade dysplastic nodules (HGDNs) and, in turn, into early-stage HCC and progress to advanced HCC [6]. Without underlying cirrhosis (20–30% cases), HCC can develop mainly on a background of HBV infection or NASH [12,13] or, less frequently, adenomas [14].

HCC arises from the accumulation of somatic mutations and epigenomic alterations. While most of them occur in ‘passenger’ genes, a few of them are regarded as ‘drivers’ responsible for the activation of key signaling pathways leading to hepatocarcinogenesis [6,12]. In dysplastic nodules and established HCC, mutations of *TERT* promoter, which encodes the synthesis of telomere reverse transcriptase, are frequent (6% in LGDNs, 20% in HGDNs, and 60% in HCC) [15]. HBV can also induce insertional mutagenesis in *TERT* promoter, as well as adeno-associated virus type 2 (AAV2), although to a much shorter extent [15,16]. The WNT-β-catenin pathway is frequently activated in HCC due to mutations in *AXIN1* and *CTNNB1* (11–37% cases) [17]. p53 inactivation and cell cycle control alterations (*CDKN2A*) are also common in HCC, especially in aflatoxin B1 exposure and HBV infection [18,19]. Furthermore, defects in chromatin remodeling complexes and epigenetic regulators are often found in HCC, including mutations in the BRG1- or HRBM-associated factors (BAFs) and polybromo-associated BAF (PBAF) chromatin complex [13,15]. 

Receptor tyrosine kinase (RAS-RAF-MAPK) and phosphatidylinositol-3-kinase, Protein kinase B and mammalian target of rapamycin (PI3K-AKT-mTOR) pathways are usually activated in HCC, owing to the amplification of regions that includes *FGF19* (5% tumors) and mutations in *RPS6KA3* and *RSK2* (5–9% cases) [16,20]. 

Oxidative stress signaling pathway is also activated through activating mutations in nuclear factor erythroid 2-related factor 2 (*NFE2L2* or NFR2) or the inactivation of Kelch-like ECH-associated protein 1 (*KEAP1*) [21]. DNA amplifications take place in chromosome regions 11q13 and 6p21, affecting the oncogene cyclin D1 (*CCND1*) and neoangiogenic vascular endothelial growth factor A (*VEGFA*) respectively, inducing the latter tumor proliferation through the secretion of macrophage-mediated hepatocyte growth factor (HGF) [22,23]. Unfortunately, most of the mutations in HCC occur in non-druggable pathways such as in the WNT-β-catenin, p53, or the *TERT* promoter, while those mutations located in more easily treatable targets are only present in a low percentage of patients, making it difficult to apply specific therapies [16,24].

#### 1.2.3. Molecular Classes

Genomic, transcriptomic, and epigenomic profiling analyses have allowed the establishment of a molecular classification of HCC. Despite the fact that this classification is not used yet in clinical practice, it correlates with clinical features [25,26]. Two molecular subtypes have been identified: the proliferation class and the nonproliferation class [27,28]. Cell proliferation and survival pathways, such as PI3K-AKT-mTOR, RAS-MAPK and MET, chromosomal instability, *TP53* inactivation, *FGF19* and *CCND1* amplifications, and α-fetoprotein overexpression characterize the proliferation class. This proliferation class is associated with HBV infection and has a poor clinical outcome [29,30]. On the other hand, tumors that belong to the nonproliferation class often have an activation of *CTNNB1* and more *TERT* promoter mutations. Transcriptionally, those tumors are similar to normal hepatocytes and are related to alcohol use and HCV infection etiologies and have better outcomes [13,31].

Tumor microenvironment (TME) is considered to play a fundamental role in all steps of carcinogenesis [32]. HCC has an inflammatory milieu due to viral hepatitis, alcohol abuse, and NAFLD or NASH. Immune cells, such as lymphocytes and macrophages, stellate cells, and endothelial cells interact with hepatocytes in the chronically inflamed liver [6,12]. According to this, HCCs that have high immune cell infiltration, activation of programmed cell death protein 1 (PD-1)/programmed cell death 1 ligand 1 (PD-L1), and activation of IFNγ signaling pathway and granzyme B and perforin 1 expression could be grouped into an ‘immune class’ and constitute 30% of tumors. Two different subclasses can be found within the ‘immune class’, an adaptive T cell response can identify the ‘active immune’ subtype, whereas the ‘exhausted subclass’ exhibits TGFβ-mediated immunosuppression and T cell exhaustion [33]. However, 25% of HCC have no immune cell infiltration.

### 1.3. Surveillance 

Patients with HCC at early stages may benefit the most from surveillance, since the symptoms caused by HCC are often detected at advanced stages of the disease and, therefore, those patients are not eligible for curative treatment [6]. Survival benefits of HCC surveillance have been shown in several publications that include mathematical models, a clinical trial, and a meta-analysis of cohort studies [34,35,36]. Surveillance could be useful for patients with cirrhosis but preserved liver function (having more of 1.5% incidence of HCC per year), as well as patients who are candidates for liver transplant [37]. Patients with chronic HBV infection have different risk of developing HCC depending on their geographic region. Age, male sex, liver fibrosis, high viral replication, genotype C, and a family history of HCC also increase such risk [38]. While patients with chronic HCV infection and fibrosis should be enrolled in a surveillance program, patients who have developed NAFLD in the absence of cirrhosis are not eligible for surveillance, since the risk of HCC is likely to be rather low [13]. However, current information is limited and additional studies will be necessary to validate HCC risk in those patients. 

Abdominal ultrasonography every six months is the preferred test for surveillance. It has a sensitivity of 60–80% and a specificity of more than 90% [39]. The most common serological tumor marker is α-fetoprotein (AFP), although its sensitivity is around 60% [40], hardly appropriate for patients’ screening. 

Several recent studies have focused on evaluating extracellular vesicles, circulating tumor cells, cell-free DNA, and non-coding RNA as novel reliable biomarkers to improve sensitivity [41]. To date, liquid biopsy, as a source of blood-base biomarkers, is believed to be a very trustworthy instrument, and some of these new non-invasive tools will undoubtedly change HCC clinical management by providing more detailed individualized decision-making in patients, including prognostic outcome [41]. 

### 1.4. Diagnosis

Diagnostic algorithms based on nodule size and detection have been described elsewhere [42,43]. Imaging techniques allow the distinction of a pattern of hyperenhancement in the arterial phase and washout in venous or delayed phases on contrast-enhanced CT or MRI, as, during the malignant transformation of hepatocytes, benign lesions receive blood supply from the portal system, while malignant nodules are supplied from the hepatic artery in patients with cirrhosis [44]. Additionally, the use of immunohistochemical markers such as glypican 3, heat shock protein 70, glutamine synthetase, and clathrin heavy chain can increase accuracy at the time of diagnosis [45].

### 1.5. Staging

Most patients with HCC have concomitant liver disease. For this reason, the prognosis evaluation must include tumor stage, the degree of liver dysfunction, and performance status [13,42], along with treatment indication [46]. Besides more generic staging systems such as tumor, node, metastasis (TNM), specific systems for liver cancer has been described such as the Cancer of the Liver Italian Program (CLIP score) or the Hong Kong Liver Cancer (HKLC) staging system [47,48]. So far, the Barcelona Clinic Liver Cancer (BCLC) algorithm is the staging system most widely applied for HCC. Since 1999, when it was first introduced, it has been updated according to clinical data [49]. This staging system quantifies tumor burden depending on the number and size of lesions and the presence/absence of macrovascular tumor invasion (Figure 1). In addition, the Child–Pugh grade assesses liver function impairment, although it has limited predictive power [43,50]. The albumin-bilirubin (ALBI) score stratifies patients across BCLC stages, but its role in clinical decision-making or stratification in trials is yet to be defined [42,43]. 

High AFP serum levels are linked to a poorer prognosis. Some studies have described that increased AFP levels can predict the risk of tumor relapse after surgical resection [51] or response to loco-regional treatment and survival in HCC [52]. Vascular endothelial growth factor (VEGF), angiopoietin 2 (Ang2), or KIT may improve prognostic prediction, but these markers are still to be implemented on the individual assessment of a specific patient [42,52].

## 2. Tumor Microenvironment in HCC

The interaction of the microenvironment with the tumor plays a relevant role in HCC pathogenesis (Figure 2). The tumor microenvironment is directly implicated in the modulation of liver fibrosis, the process of hepatocarcinogenesis, the epithelial-mesenchymal transition (EMT), invasion, and metastasis [53,54].

### 2.1. Hepatic Stellate Cells

Hepatic stellate cells (HSCs) are major components of liver connective tissue. They are localized in the basolateral surface of hepatocytes and the anti-luminal side of sinusoidal cells [55]. HSCs are in charge of vitamin A storage, synthesis of matrix metalloproteinases (MMPs) and extracellular matrix components (ECM, collagen), release of cytokines (IL-6 and IL-1β), defensin-1, chemokines (CCL5, CCL2), and growth factors (TGF-α/β, EGF, PDGF, bFGF) [55,56]. Normally, HSCs are in a quiescent state. Upon liver injury, they become activated, their cytoskeleton becomes remodeled through an increased expression of alpha-smooth muscle actin (α-SMA), and there is also a rise in cytokines, ECM components, and growth factors production [55]. In the activated state, HSCs transdifferentiate into myofibroblast-like cells. This phenotype makes them more contractile, so they can infiltrate the HCC stroma and localize around fibrous septa, sinusoids, and capsules [57,58]. 

Conditioned media from tumoral hepatocytes has been found to increase the proliferation of rat HSCs and induce the expression of HSCs’ activation markers [59,60]. Similarly, another study demonstrated that collected media from HSCs potentiated the tumorigenic capacity of HCC cancer cell lines [61]. The co-culture of hepatoma cells and activated HSCs also revealed the activation of genes related to inflammation, chemotaxis, angiogenesis, and metalloproteinase from microarray analysis data [62,63]. Regarding in vivo studies, the co-implantation of HCC and HSCs cells in nude mice increased tumor growth via NF-κB and extracellular signal regulated kinase (ERK) pathways activation [61,64]. In this sense, previous work has showed that angiogenin was responsible for the crosstalk between HCC and HSCs cells both in vitro and in mice models [65].

HSCs are also involved in the promotion of angiogenesis in HCC. Diverse mechanisms are responsible for this, among them the secretion of angiopoietin-1 [66] or IL-8 [67]. Moreover, PDGF secreted by tumor and endothelial cells has been described as attracting HSCs, while at the same time, HSCs secrete VEGF, thus promoting angiogenesis [68].

Several studies have pointed out that the secretion of IL-6 by HSCs may promote HCC progression [69,70]. In an HCC murine model with obesity, insulin resistance, and dyslipidemia, fatty acid binding protein 4 (FABP4) was enriched in intra-tumoral HSCs, contributing to hepatocarcinogenesis [71]. The co-culture of HSCs with HCC cells demonstrated that the overexpression of miR-1246 secreted by HSCs or the silencing of its target RORα increased proliferation, invasion, and metastasis of HCC cells, with the involvement of the Wnt/β-catenin pathway [72].

HSCs have also been described as promoting tumor chemoresistance. The laminin-332/α3 integrin axis and the ubiquitination of focal adhesion kinase (FAK) by HSCs were demonstrated to be involved in sorafenib chemoresistance [73]. In addition, FGF9, expressed only by HSCs, promoted the tumorigenic capacity of HCC cells and the resistance to sorafenib, and FGF9 overexpression was associated with poor prognosis in patients with HCC [74].

While most studies favor a role for HSCs in promoting HCC, HSCs have also been found to delay HCC progression. In particular, endosialin secreted by HSCs was reported to negatively regulate HCC proliferation in inducible mouse models of HCC [75].

### 2.2. Cancer-Associated Fibroblasts

Fibroblasts are present in the fibrillar matrix of connective tissue. They are responsible for wound healing, formation of extracellular matrix (ECM), tissue maturation, and the inflammatory response [76]. Cancer-associated fibroblasts (CAFs) are a sub-group of fibroblasts that are activated and implicated in cancer progression. Although CAFs arise from normal fibroblasts, CAFs can also derive from epithelial cells, endothelial cells, smooth muscle cells, bone marrow-derived progenitor cells, and pre-adipocytes [77]. Additionally, HCC tumors frequently develop in a cirrhotic liver in which there is a great amount of activated fibroblasts [78]. Therefore, CAFs may contribute to HCC tumor progression by producing growth factors (EGF, FGF, HGF, and TGF-β), chemokines (SDF-1), cytokines (IL-6), and metalloproteinases (MMP-3 and MMP-9) [79,80,81]. Moreover, the exosomal miR-1228-3p released by CAFs and directed to HCC cells was described as involved in chemoresistance [82]. In this regard, there is a growing amount of evidence showing that the crosstalk between CAFs and HCC tumors could be mediated by miRNAs contained in exosomes. For example, low miR-150-3p levels secreted by CAFs have been discovered to be involved in HCC migration and invasiveness as well as poor clinical outcome [83]. Interestingly, the upregulation of mirR-335-5p by CAFs inhibited HCC tumor cells proliferation in vitro and in vivo [84]. Moreover, HCC tumor cells were found to induce the conversion of HSCs into CAFs through the secretion of miR-21, which promoted cancer progression via the secretion of the angiogenic factors VEGF, MMP2, MMP9, bFGF and TGF-β [85].

### 2.3. Tumor-Associated Macrophages

Macrophages around the tumor site are called tumor-associated macrophages (TAMs). Macrophages can display the M1 (classic) or M2 (alternative) phenotype depending on their tumor-suppressing or tumor-promoting role [86]. M1 macrophages produce Th1-cytokines, such as IFN-γ, and are activated by LPS and other microbial antigens. They exhibit high antigen-presenting capacity and increased cytotoxic activity, thereby producing reactive oxygen species (ROS) [87]. On the contrary, M2 macrophages are polarized by Th2-type cytokines IL-4, IL-13, glucocorticoids, and TGF-β. Their antigen-presenting capacity is low. M2 macrophages decrease inflammation, suppress the adaptive immune system, and promote tumor progression, angiogenesis, and tissue repair [88].

In HCC, M2 macrophages have been found to promote tumor progression and metastasis with the involvement of glypican-3, a member of the glypican family of heparin-sulfate proteoglycans reported to be highly expressed in the majority (>70%) of HCCs [89]. In addition, TGF-β1 secretion by TAMs promoted cancer progression and EMT in HCC [90,91], and moreover, the TAM-production of IL-6, via STAT3, also promoted stemness in HCC [92]. Moreover, in a murine model of HCC, intra-tumoral macrophages expressing MMP-9 were involved in ECM remodeling, thus favoring tumor progression [93], while, in another study, the presence of TAMs correlated with tumor vascularity, pointing towards the ability of TAMs to promote angiogenesis [94].

It has been shown, in Hepa1-6 HCC tumors, that, in the early phase of tumor development, infiltrated macrophages displayed a tumor-suppressing phenotype, while, at advanced stages, the TAM population increases and is associated with tumor progression [95]. Thus, it is becoming apparent that macrophage polarization plays a crucial role in the initiation of liver diseases, and its role in HCC needs to be further clarified, particularly since it may affect immunotherapy efficacy [96]. At the same time, tumor cells have been found to release Wnt ligands that promoted M2 polarization of macrophages and, in turn, promoted tumor growth, invasion, and immunosuppression in HCC [97]. In this regard, treatment of HCC with sorafenib has been shown to induce the repolarization of alternative macrophages to M1 phenotype through IGF-1 signaling [98]. 

Additionally, in HCC human samples, TAM infiltration was linked with PD-L1 overexpression [99]. Although M1 macrophages have been considered to exert an anti-tumor role, M1 macrophages my promote PD-L1 expression in HCC tumor cells, highlighting the potential role of M1 macrophages in tumor promotion through IL-1β pathway [100]. In fact, Kupffer cells, resident macrophages in the liver, have been reported to mediate tumor growth in HCC by producing PD-L1 that interacts with PD-1 receptor in CD8^+^ T cells, impairing CD8^+^ T cell response [101]. In addition, Kupffer cells produce osteopontin, which is involved in inflammation, tumor progression, and metastasis [102]. 

### 2.4. Endothelial Cells

Endothelial cells (ECs) are present in the interior face of blood vessels. Other cells, such as HSCs, participate in controlling the size and elasticity of liver vessels [103]. The interactions of ECs with the ECM and basement membrane proteins play a role in proliferation, stability, and neoangiogenesis. When the basement membrane degrades, ECs become exposed to collagen, which triggers the formation of new blood vessels [104]. Neovascularization favors tumor proliferation, invasion, and metastasis, since the new blood supply provides oxygen and nutrients to the tumor [105]. Tumor blood vessels have structural abnormality and increased permeability. ECs carry angiogenic receptors, for instance VEGFR, EGFR, PDGFR, and CXCR [106]. Additionally, hypoxia is a known driver of tumor angiogenesis. Many studies conducted in HCC preclinical models have shown that hypoxia-inducible factor (HIF) proteins led to the activation of VEGF, which promotes angiogenesis [107,108,109]. VEGF and VEGFRs are crucial for HCC development [110,111]. The binding of VEGF ligands to their receptors elicits downstream phosphorylation that results in EC proliferation and the formation of new branches of blood vessels [112]. High VEGF levels in serum have been found to be associated with bad prognosis in HCC patients who underwent surgical resection [113], since sVEGF concentration has been showed to correlate with angiogenesis, invasion, and metastasis of HCC [114]. The interaction of platelet-derived growth factors (PDGF) with PDGF receptors (PDGFR) triggers the activation of the same signaling pathways as the binding of VEGF and VEGFRs not only in ECs but also in fibroblasts, smooth muscle cells, and HSCs [115]. In this sense, PDGFRα expression was associated with microvascular invasion [116]. 

Additionally, fibroblast growth factor (FGF) and fibroblast growth factor receptors (FGFR) also regulate cell growth and angiogenesis [117]. Basic fibroblast growth factor (bFGF) fostered VEGF expression and its synergistic effect contributed to HCC development and neovascularization [118]. Of interest, angiopoietin-1 (Ang-1) and 2 (Ang-2) bind to their receptor, Tie2, to stimulate angiogenesis [119]. Ang-1 and Ang-2 expression was detected in hepatoma, HSCs, ECs, and smooth muscle cells, while Tie2 receptor was only identified in ECs, HSCs, smooth muscle cells, and monocytes [120,121]. Ang-2 serum levels were high in patients with cirrhosis and HCC [122], being a prognosis marker [123]. Ang-2 exhibited a synergistic effect with VEGF in the development of angiogenesis in HCC in mice through the activation of MMP-2 and MMP-9 [124]. Ang-2 was included in a five-gene signature that effectively predicted HCC rapid growth [125]. As other pro-angiogenic factors, Ang-2 also played a role in the promotion of HCC invasion and metastasis [126].

### 2.5. Tumor-Associated Cells of the Innate Immune System

Innate immune mechanisms may support or neutralize tumor-related immune activation, being recognized drivers of disease progression in the liver, particularly during conditions such as fibrosis or cirrhosis prior to HCC. Exhaustive research has been developed to delineate the immunological steps involved in the initiation and evolution of liver cancer. During HCC progression, several studies analyzing the response to immunotherapy have leaded to conflicting results, probably due to the complex and only partially known interactions between specific immune cells, tumor cells, and the different cells that configure the tumor microenvironment. For reviews on the subject, see [127,128].

Forgotten during years, tumor-infiltrating immune cells in the HCC have been recently evaluated and characterized [128]. For many solid tumors, including HCC, different relationships between immune cell populations and therapy efficacy and prognosis have been suggested. While the complete impact of the tumor immune environment is still to be determined, myeloid cells including TAMs and myeloid-derived suppressor cells (MDSCs) are abundantly present in the HCC microenvironment being frequently associated with poor prognosis. In general, myeloid cells in HCC play a very active role in promoting tumor initiation, development, angiogenesis, metastasis, and even therapeutic resistance [129]. In contrast, increasing numbers of infiltrating T-effector cells are habitually linked with a good prognosis [130]. Generally, a pro-inflammatory HCC ambient with infiltrating natural killer (NK) cells, and CD8-expressing T cells are considered to be positive and associated with good clinical outcomes in numerous tumor types [131]. NK cells play a central role in hepatic immunity, accounting for 25–50% of the total number of liver lymphocytes. Both circulating and tumor infiltrating NK cells are positively correlated with patient survival benefit in HCC [132], contrary to other immune cells, such as MDSCs and regulatory T cells, which seem to disrupt the immune control of the HCC [130].

## 3. Systemic Therapies for HCC

Clinical treatment of HCC includes surgical therapies, resection or tumor ablation, transplantation, transarterial chemoembolization (TACE), therapies that have been extensively revised [13,42,43].

HCC patients with a single tumor and preserved liver function are candidates for resection. Liver transplantation benefits patients who are not good candidates for surgical resection and who present with a solitary tumor ≤5 cm or up to three nodules ≤3 cm. Image-guided ablation is the most frequently used therapeutic strategy, but its efficacy is limited by the size of the tumor and its localization. TACE has survival benefit in asymptomatic patients with multifocal disease without vascular invasion or extrahepatic spread [42]. Finally, systemic therapies are only recommended in advanced HCC and with well-preserved liver function [42]. No systemic drugs were available for patients with advanced stage HCC until 2007, when sorafenib was approved [43]. Sorafenib increased the available treatment options for patients with extrahepatic spread and vascular invasion and improved survival in patients with advanced HCC.

Nonetheless, various limitations such as low response rates, resistance to sorafenib, or adverse effects (e.g., hand–foot skin reaction) prompted concerted efforts aimed at developing new molecular targeted agents to provide more treatment options and second-line agents for patients with disease progression or intolerance to sorafenib.

Of importance, during the past five years, many trials have been made in the search for novel and more effective systemic treatments for advanced HCC, not only as first-line but also as second-line, as recently reviewed in an EASL position paper aimed at helping clinicians provide the best possible care for patients today [133]. Therefore, as of today, drugs licensed in some countries include four oral multi-tyrosine kinase inhibitors (MKIs) (sorafenib, lenvatinib, regorafenib, and cabozantinib), one anti-angiogenic antibody (ramucirumab), and four immune checkpoint inhibitors, alone or in combination (atezolizumab in combination with bevacizumab, ipilimumab in combination with nivolumab, nivolumab and pembrolizumab in monotherapy) (Figure 3).

According to this updated guidelines, in the next paragraph we will introduce the systemic treatments approved in recommended order of use.

### 3.1. First-Line Therapies

#### 3.1.1. Atezolizumab-Bevacizumab (Atezo-Bev)

To date, atezolizumab and bevacizumab combination is the first treatment superior to sorafenib demonstrating prolonged overall survival (67.2% vs. 54.6%; hazard ratio [HR] 0.58) and progression-free survival (6.8 months vs. 4.3 months; HR 0.59) [134]. The success of IMbrave 150 clinical trial has changed the paradigm of HCC treatment, and atezo-bev has become the recommended systemic therapy if no contraindications are present [133].

Atezolizumab (Tecentriq) is a humanized IgG1 monoclonal antibody that targets PD-L1 to prevent its binding with PD-1 and B7-1 receptors, thus reversing T-cell suppression [135]. Bevacizumab (Avastin) is a monoclonal antibody that targets vascular endothelial growth factor (VEGF), inhibiting angiogenesis and tumor growth [110]. Anti-VEGF therapy also enhances anti-PD-1/PD-L1 activity by reducing VEGF-mediated immunosuppression and promoting T-cell infiltration in tumors [136]. Of note, other immune checkpoint inhibitors [137,138], as well as atezolizumab [139], in monotherapy, did not reach better outcome in HCC patients, highlighting the synergistic efficacy of immunotherapy and anti-angiogenic combination.

Regarding adverse effects, hypertension and increased AST or ALT are grade 3 or 4 adverse events frequently observed after atezo–bev treatment. Moreover, bleeding, a life-threatening risk for cirrhotic patients, is a common reaction to bevacizumab. In this sense, risk of bleeding, comorbidities such as arterial hypertension and cardiovascular disease, and prior autoimmune conditions may become limiting parameters for the indication of atezo-bev. If the patient has contraindications to atezo-bev, alternative therapies should be considered, such as sorafenib or lenvatinib.

Another immune-based therapy that will likely be included in the first line soon is the tremelimumab/durvalumab combination. Although the peer review data have not yet been published, a significant survival benefit over sorafenib has been announced in the HIMALAYA phase III trial. Once the study data are fully available, changes in clinical decision-making are expected in such a setting, although these are now difficult to foresee.

#### 3.1.2. Sorafenib

Sorafenib (Nexavar) is a small molecule that inhibits the phosphorylation of up to 40 tyrosine kinases, including VEGFR1, 2 and 3, PDGFRβ, KIT, and RET. This tyrosine kinase inhibitor (TKI) also suppresses Raf kinase isoforms, such as wild-type Raf1, B-Raf, and mutant b-raf V600E. Sorafenib displayed anti-proliferative, anti-angiogenic, and pro-apoptotic properties in HCC cell lines [140], anti-tumor activity in tumor xenograft nude mice [141], and anti-metastatic effect in preventing postsurgical recurrence in an orthotopic mouse model [142]. The efficacy of sorafenib possibly lays in its capacity to target both tumor cells and their microenvironment [6]. As an example, it has been described that sorafenib also had an impact on HSCs proliferation by the suppression of α-SMA and PDGF-related pathways, which decreased HCC cell viability [143]. However, a high dose of sorafenib has been described as promoting immunosuppression through the induction of PD-1 expression in infiltrating immune cells in a murine HCC model [144]; whether this could affect patients, particularly those under immunotherapy, is an aspect that deserves to be studied.

Sorafenib was the first compound that demonstrated survival benefit in HCC in a phase 3, double-blind trial versus placebo (SHARP trial). The median overall survival for patients in the sorafenib arm was 10.7 months compared to 7.9 months in the control group (HR 0.69, 95% confidence interval 0.55–0.87, *p* < 0.001) [145]. In a parallel trial conducted in the Asian-Pacific population, sorafenib showed a similar survival benefit [146]. The most common adverse effects are diarrhea (8–9% patients) and a hand–foot skin reaction (8–16% patients) [145]. Sorafenib is recommended as the standard systemic therapy for HCC in the first line setting in patients with well-preserved liver function (Child–Pugh A or early B class), with advanced tumors, BCLC-C, or tumors that progressed after loco-regional therapies [43]. The appearance of dermatologic reactions has been linked to better survival following sorafenib administration [147].

Among the molecular mechanisms responsible for sorafenib effectivity in HCC cells is the activation of programmed cell death, apoptosis, provoked by the downregulation of myeloid cell leukemia sequence 1 (MCL-1) expression, an anti-apoptotic member of the BCL-2 family [148]. Recent data have shown that the mitochondrial link with sorafenib activity is more profound. Sorafenib induces mitochondrial reactive oxygen species (ROS), depletes mitochondrial membrane potential, and induces changes in the BCL-2/MCL-1 ratio [149,150]. In fact, continuous sorafenib exposure altered the levels of anti-apoptotic BCL-2 proteins allowing HCC cell death escape. In contrast, surviving cells are sensitized against BH3-mimetics, inhibitors of specific BCL-2 proteins such as navitoclax [150]. Sorafenib has also been described as involved in the autophagy pathway. The administration of autophagy inhibitors, such as chloroquine or pemetrexed, improved sorafenib efficacy in tumor cells and nude mice hepatoma tumors [151]. Additionally, MCL-1 downregulation was found to disrupt the MCL-1:Beclin 1 complex and induce autophagic cell death in HCC cell lines [152]. In fact, as a consequence of the mitochondrial damage induced by sorafenib, mitophagy is also activated by a triggering mechanism that involves mitochondrial ROS production [153], allowing sorafenib activity to be modulated by antioxidant administration [154]. Acquired drug resistance, which reduces sorafenib effectiveness in patients, may depend on these or other mechanisms. HCC is highly heterogeneous, within the tumor and among individuals, and this influences disease progression, classification, prognosis, and, naturally, cellular susceptibility to drug resistance. In this sense, long-term exposure to sorafenib of hepatoma cells provoked the acquisition of chemoresistance, as well as EMT features [155,156]. Hypoxia has been described to be involved in sorafenib resistance due to HIF-1α and NF-κB activation [157]. Moreover, M2 macrophages have been found to participate in sorafenib resistance by the release of HGF [158].

#### 3.1.3. Lenvatinib

Lenvatinib (Lenvima) is an inhibitor of VEGFRs, RET, KIT, PDGFRα, and FGFR1-FGFR4 [159]. It also displayed anti-angiogenic properties and anti-FGFRs activity in hepatoma cells and xenografts [160,161]. Lenvatinib has been described to exert an immunomodulatory effect through the increase of CD8^+^ T cell population while diminishing macrophages and monocytes populations in HCC cells [162].

In a phase 3 clinical trial, lenvatinib showed to be non-inferior to sorafenib in terms of overall survival. Hypertension, diarrhea, or a decrease in appetite or weight were among the most common adverse events [163]. In a small group of patients, the levels of AFP were found to decrease in the next two weeks following treatment, suggesting that AFP levels could be predictive of patients’ response [164]. Furthermore, circulating FGF-19 and Ang-2 have been proposed as predictors of clinical response to lenvatinib in HCC patients [165,166], as well as an early tumor shrinkage [167]. However, like sorafenib, HCC has been described as displaying resistance against lenvatinib. The HGF/c-MET signaling activation was identified as one mechanism of lenvatinib tolerance [168].

### 3.2. Second-Line Therapies

#### 3.2.1. Regorafenib

Regorafenib (Stivarga) is a multikinase inhibitor (MKI) against VEGFR-2, VEGFR-3, KIT, RET, wild-type, and mutant (V600E) B-Raf, PDGFR, FGFR1, angiopoietin 1 receptor (TIE2), RET, and p-38-alpha. Its inhibitory profile is slightly different from sorafenib, since regorafenib has stronger potency targeting VEGFR and TIE2, KIT, and RET [169]. Like sorafenib, regorafenib inhibits angiogenesis, oncogenesis, and tumor microenvironment. Regorafenib was shown to block cell growth and invasion in hepatoma cell lines [170]. This MKI also targeted MAPK pathway, induced caspase cleavage and activated the autophagic pathway [171,172], and mitophagy as a consequence of its mitochondrial activity [154]. In fact, regorafenib alteration of mitochondrial proteins such as BCL-xL is related to regorafenib resistance, pointing to BH3 mimetics for combined therapies [173]. Moreover, both intrinsic and extrinsic apoptotic pathways were activated by regorafenib [174]. The treatment with regorafenib provoked a decrease in the expression of metastasis-related proteins in HCC cells [175]. Regorafenib was demonstrated to block EMT activation and overcome the acquired resistance to sorafenib [176].

The RESORCE trial was the first phase 3 clinical trial that showed that patients who progressed on sorafenib benefited from oral regorafenib administration versus placebo in a second line setting [177]. Median survival was 10.6 months for the regorafenib arm, while 7.8 months for the control group (HR 0.63; 95% 0.50–0.79; *p* < 0.0001). Manageable adverse events consisted of a hand–foot skin reaction, hypertension, and fatigue. Additional analyses of the RESORCE trial have suggested that the administration of regorafenib following sorafenib may extend survival [178].

#### 3.2.2. Cabozantinib

Cabozantinib (Cometriq, Cabometyx) is a small molecule with tyrosine kinase inhibitory prolife against VEGFR-2, RET, KIT, FLT-3, TIE2, and AXL. Cabozantinib differs from sorafenib and regorafenib in that it is capable to also block c-Met [179]. Cabozantinib has demonstrated anti-tumor activity in HCC cells by inhibiting tumor growth, angiogenesis, invasion, and migration. It also reduced the number of HCC metastatic nodules in the lungs and liver in mice [180]. In a phase two clinical trial, cabozantinib demonstrated effectivity in HCC patients [181]. Those promising results led to the conduction of a phase 3 clinical trial in patients who progressed after sorafenib treatment. Cabozantinib increased overall survival (10.2 months) compared to placebo (8.0 months, HR 0.76; 95% CI, 0.63–0.92; *p* = 0.005). The most frequent side effects were palmar-plantar erythrodysesthesia, hypertension, increase AST, fatigue, and diarrhea [182].

#### 3.2.3. Nivolumab

Nivolumab (Opdivo) is a human monoclonal antibody that targets programmed cell death protein 1 (PD-1). It is an immune checkpoint inhibitor, since nivolumab impedes the signaling that blocks T cell anti-tumor activity [183]. A phase 1/2 dose escalation study performed with advanced HCC with or without previous sorafenib treatment showed the potential of nivolumab for the treatment of HCC (CheckMate 040 trial) [184]. A further analysis of the CheckMate 040 trial highlighted that some inflammatory biomarkers trended with improved survival and an anti-tumor immune response [185]. Nevertheless, a subset of patients with hyperprogressive disease (HPD) was identified after nivolumab treatment in HCC patients [186]. Furthermore, administration of nivolumab plus ipilimumab, which targets CTLA-4, a inhibitory T-cell receptor, also showed to be a promising therapeutic strategy in HCC patients who progressed on sorafenib [187].

#### 3.2.4. Pembrolizumab

The humanized monoclonal antibody pembrolizumab (Keytruda) blocks PD-1 as well. In a non-randomized phase 2 clinical trial, pembrolizumab was effective in patients who were treated previously with sorafenib (KEYNOTE-224) [188]. These results led to testing pembrolizumab compared to placebo in a phase 3 randomized clinical trial. Although median overall survival was longer for the pembrolizumab arm, 13.9 months (95% CI, 11.6 to 16.0 months) and 10.6 months (95% CI, 8.3 to 13.5 months) for placebo, the results were not statistically significant [137].

#### 3.2.5. Ramucirumab

Regarding antiangiogenic therapies, ramucirumab (Cyramza), a monoclonal antibody against VEGFR2 [189,190,191], failed to improve survival in the REACH trial in patients treated previously with sorafenib. However, the authors identified AFP serum levels as a prognostic marker showing that patients with high levels of AFP (≥400 ng/mL) benefit from ramucirumab treatment. These observations were validated in REACH-2, a double-blind phase III trial, wherein only patients treated with sorafenib with high AFP levels were included. Ramucirumab improved overall survival (8.5 versus 7.3 months HR 0.710, 95% CI 0.531–0.949; *p* = 0.0199) and has become the first HCC therapy with biomarker-guided patient selection. Hypertension, liver failure, and hyponatremia were the most common grade 3–4 adverse events.

#### 3.2.6. Combination Therapies

Regarding ongoing clinical studies, several combinations of treatment regimens are being tested in patients with HCC in both the first line and second line: the RENOBATE study (combination of regorafenib and nivolumab administered as first-line therapy in unresectable HCC), the REGOMUNE trial (avelumab, which targets PD-L1, will be studied together with regorafenib), the GOING trial (second-line treatment with regorafenib, followed by nivolumab treatment in patients who have progressed on sorafenib administration), the ACTION trial (will evaluate the effectivity of cabozantinib in patients who are sorafenib-intolerant or who do not meet the RESORCE criteria), and the COSMIC-312 clinical trial (administration of cabozantinib in combination with the immune checkpoint inhibitor atezolizumab), among some others.

## 4. Conclusions

After years without major therapeutic improvements and with increasing incidence, finally advances are arriving for HCC treatment. Sorafenib is no longer the only systemic therapy for patients, and novel combinations are already working in clinical trials. Accumulating data demonstrate that etiology and the HCC microenvironment have a major influence on tumor growth and immune control. The improved knowledge of the specific molecular mechanisms involved is expected to provide evidence-based information critical for clinical management. Based on that, novel recommendations for treatment are already being suggested that should help physicians [192]. A diligent effort from translational researchers is required to provide tools to facilitate decision-making, and careful analysis of the novel therapeutic strategies will be necessary to ensure rapid benefit for HCC patients. Important steps to revert the dismal prognosis associated with HCC have been made, and now is time to decide the next ones and to guarantee their correct implementation.

## Figures and Tables

**Figure 1 cancers-14-00621-f001:**
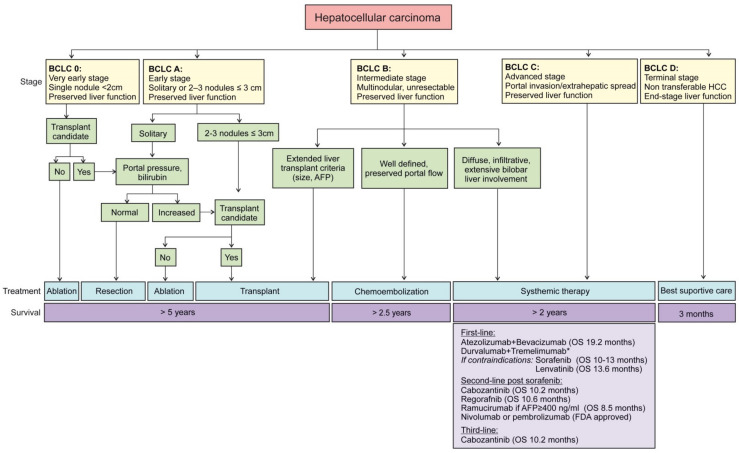
Updated treatment strategy in HCC management. The Barcelona Clinic Liver Cancer (BCLC) staging recommends HCC treatment in accordance with five defined stages. Local curative treatments including resection, ablation, or transplantation are endorsed for asymptomatic patients with preserved liver function and low tumor burden. Systemic therapies should be applied to patients in advanced stage or even in intermediate stage, when transplantation is not an option and chemoembolization not recommendable due to the presence of portal hypertension or the number/location of nodules. Current systemic therapies are presented. * Not yet FDA-approved, positive Phase III trial report.

**Figure 2 cancers-14-00621-f002:**
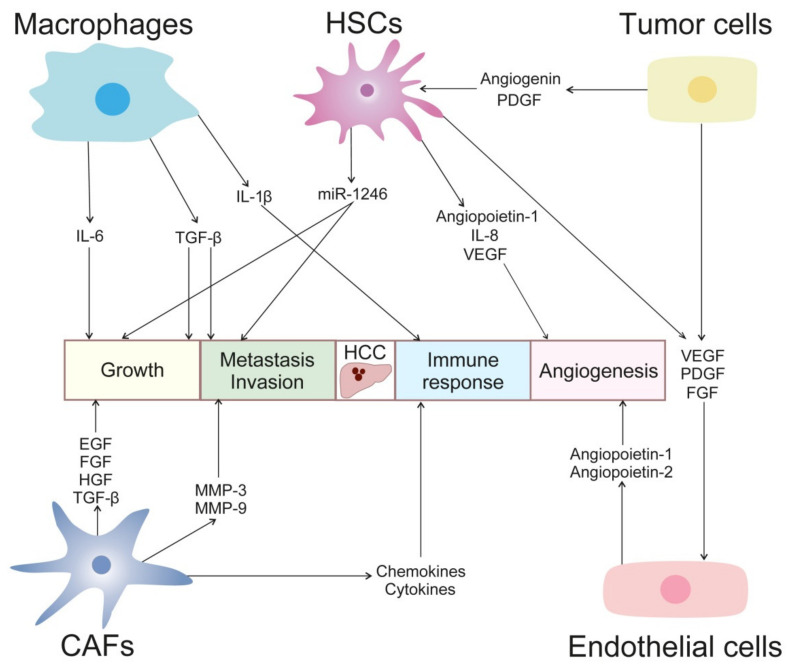
Cellular cross-talk in HCC development. Tumor microenvironment plays a critical role in HCC progression. Growth factors, cytokines, chemokines, metalloproteinases, miRNAs, and angiogenic factors mediate crosstalk between tumor, endothelial and stellate cells, fibroblast, macrophages, and other immune cells. These interactions promote tumor growth, neovascularization, invasion, and immunosuppression.

**Figure 3 cancers-14-00621-f003:**
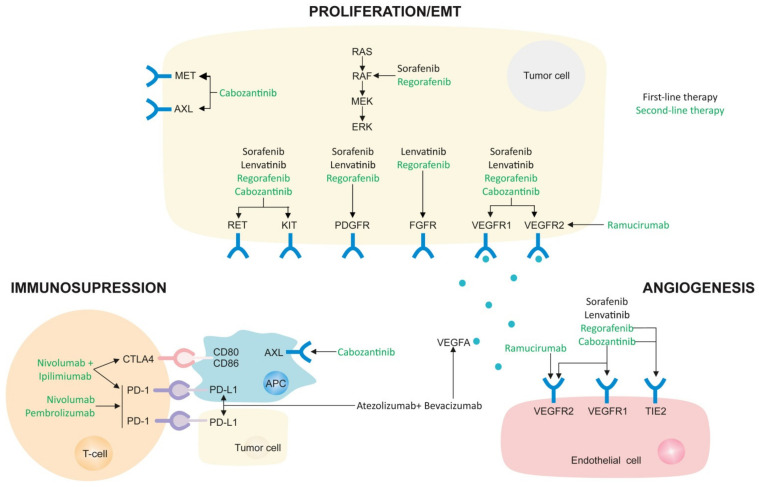
The mechanisms of action of currently approved molecular therapies. Main targets are indicated for each drug and separated depending on the activity against proliferation/EMT with multikinase inhibitors, decreasing angiogenesis in tumor microenvironment, or blocking tumor immunosuppression with checkpoint inhibitors.

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
