# Peer review of "Hepatocellular Carcinoma: Molecular Pathogenesis and Therapeutic Advances"

_cancers, 2022, doi:10.3390/cancers14030621_

Round 1

Reviewer 1 Report

In the paper entitled “Hepatocellular Carcinoma: Molecular Pathogenesis and Therapeutic Advances” the authors summarize the key cellular and molecular aspects characterizing hepatocellular carcinoma (HCC) and available therapeutic options through an extensive analysis of current literature. In general, the paper is well written, and many references are quoted. However, some caution should be used when literature data are reported. Moreover, in the abstract the authors mention the need of novel tissue and serum biomarkers (line 48-49) but then this issue is not examined thereafter.

A point of major concern is at page 7, line 300-301, in which a paper (reference 96) is quoted. Such published paper, which has been retracted (Hepatology, VOL. 74, NO. 6, 2021; DOI: 10.1002/hep.32229), presents several problems besides those, which caused retractation.

Minor points of concern:

  • Page 7, line 311, “by producing PD-1 receptor that interacts with PD-L1 in CD8+ T cells” must be changed as follow: “by producing PD-L1 that interacts with PD-1 receptor in CD8+ T cells”.
  • Page 7, line 316, the following part of the sentence should be rephrased “that they conform and control the vessel size and elasticity”.
  • Page 8, line 347, please change “Ang-2 was also played” a follow: “Ang-2 also played”.
  • Page 10, line 437, “It has been shown to target MAPK pathway as well”. This sentence is referred to the activity of sarafenib but we know the drug targets RAF, which is a component of MAPK pathway. So, does it also target other kinases, which are downstream to RAF? If so, please include the appropriate reference.
  • Page 10, lines 444-445, “Moreover, sorafenib was described to promote immunosuppression through the induction of PD-L1 expression in infiltrating immune cells [144]” Please change PD-L1 to PD-1 as reported in reference 144. If the drug promotes immunosuppression this is not good news for patients. Since this effect seems to be concentration dependent how does it correlate to standard therapeutic doses?
  • The simple summary needs some language improvement.

Author Response

In the paper entitled “Hepatocellular Carcinoma: Molecular Pathogenesis and Therapeutic Advances” the authors summarize the key cellular and molecular aspects characterizing hepatocellular carcinoma (HCC) and available therapeutic options through an extensive analysis of current literature. In general, the paper is well written, and many references are quoted.

However, some caution should be used when literature data are reported. Moreover, in the abstract the authors mention the need of novel tissue and serum biomarkers (line 48-49) but then this issue is not examined thereafter.

We have removed this line and modified the abstract to be more related to the content of the review. Thank you.

A point of major concern is at page 7, line 300-301, in which a paper (reference 96) is quoted. Such published paper, which has been retracted (Hepatology, VOL. 74, NO. 6, 2021; DOI: 10.1002/hep.32229), presents several problems besides those, which caused retractation.

We thank the reviewer for pointing this out. The article was retracted last month, after we finished writing our review, and no one noticed, except the reviewer of course. Thanks again.

Minor points of concern:

Page 7, line 311, “by producing PD-1 receptor that interacts with PD-L1 in CD8+ T cells” must be changed as follow: “by producing PD-L1 that interacts with PD-1 receptor in CD8+ T cells”.

It has been changed.

Page 7, line 316, the following part of the sentence should be rephrased “that they conform and control the vessel size and elasticity”.

It has been rephrased.

Page 8, line 347, please change “Ang-2 was also played” a follow: “Ang-2 also played”.

Done.

Page 10, line 437, “It has been shown to target MAPK pathway as well”. This sentence is referred to the activity of sarafenib but we know the drug targets RAF, which is a component of MAPK pathway. So, does it also target other kinases, which are downstream to RAF? If so, please include the appropriate reference.

This confusing remark has been deleted.

Page 10, lines 444-445, “Moreover, sorafenib was described to promote immunosuppression through the induction of PD-L1 expression in infiltrating immune cells [144]” Please change PD-L1 to PD-1 as reported in reference 144. If the drug promotes immunosuppression this is not good news for patients. Since this effect seems to be concentration dependent how does it correlate to standard therapeutic doses?

This observation has been modified and better explained.

The simple summary needs some language improvement.

Certainly. We have completely rewritten the simple abstract to better explain the content of our manuscript. Thanks for suggesting this.

Thank you very much for your time and nice review.

Reviewer 2 Report

The authors have clearly demonstrated some critical aspects of HCC molecular pathogenesis in this review. Also, they provide updated information for the treatment for HCC, including first-line and second-line therapies. The content in this review is sufficient. Some suggestions are recommended for the authors:

  1. In lines 121-122, the authors mentioned some mutation is non-druggable. I will suggest the authors create a table to indicate which mutations on the corresponding gene are druggable and which are not. Also, the author may mention the drug to be used for dealing with the druggable target. It will help the readers digest the content.
  2. Section 2 is a bit complicated and involves a lot of concepts. A diagram will be recommended to illustrate how tumour-associated fibroblast/macrophage, hepatic stellate cell, endothelial cells, and immune cells contribute to HCC development.

Author Response

The authors have clearly demonstrated some critical aspects of HCC molecular pathogenesis in this review. Also, they provide updated information for the treatment for HCC, including first-line and second-line therapies. The content in this review is sufficient. Some suggestions are recommended for the authors:

In lines 121-122, the authors mentioned some mutation is non-druggable. I will suggest the authors create a table to indicate which mutations on the corresponding gene are druggable and which are not. Also, the author may mention the drug to be used for dealing with the druggable target. It will help the readers digest the content.

These lines have been changed. There is a new paragraph to better explain this point. We have not included a table with pharmacological mutations since there are many of them, with a low percentage of patients each, and we are not sure if this could mislead readers by providing information that is not very relevant to the current treatment of HCC.

Section 2 is a bit complicated and involves a lot of concepts. A diagram will be recommended to illustrate how tumour-associated fibroblast/macrophage, hepatic stellate cell, endothelial cells, and immune cells contribute to HCC development.

Following the reviewer's suggestion, we have included an additional figure in our manuscript to illustrate the cellular crosstalk established during HCC development. Thanks for your recommendation.

Thank you very much for your time and nice review.

Round 2

Reviewer 1 Report

The manuscript was revised by authors according to suggestions